# Multicriteria Model to Support the Hiring of Road Freight Transport Services in Brazil

**Eduardo Carvalho Moretto \* and Aldery Silveira Júnior \***

Department of Business Management, Faculty of Economics, Administration, Accounting and Public Policy Management, University of Brasilia, Brasilia 70910-900, DF, Brazil

\* Correspondence: eduardomorettopvh@gmail.com (E.C.M.); aldery@unb.br (A.S.J.)

**Abstract:** The transportation of goods plays a fundamental role in the global economy. In Brazil, specifically, a significant portion of what is transported goes through highways, and the provision of this service is carried out through the hiring of specialized companies or independent drivers, who end up serving companies from various sectors. This study successfully constructed a decision model to support the hiring of road freight transport services in Brazil. To achieve this, the multicriteria decision analysis (MCDA) approach was used, employing the Measuring Attractiveness By a Categorical-Based Evaluation Technique (MACBETH) method. The development of this model was informed by a comprehensive literature review, interviews with three transportation professionals, and a simulation involving eight anonymous Brazilian companies. This initiative aims to create a practical framework for effectively selecting logistics operators in the road transport sector to meet the needs of companies dependent on these services.

**Keywords:** freight transport; multicriteria model; road transport; transport hiring; MACBETH

## 1. Introduction

Road freight transport plays a fundamental role in the economy of Brazil, being responsible for the movement of the majority of products and goods throughout the national territory [1].

The objective of this study was to create a decision model to facilitate the hiring of transportation services. To achieve this, the study employed the multicriteria decision analysis (MCDA) methodology along with the Measuring Attractiveness By a Categorical-Based Evaluation Technique (MACBETH) method. The study evaluated proposals from eight anonymous companies for transporting three tons of soybeans on the São Paulo (SP) to Belo Horizonte (MG) route.

The transportation of goods dates back to the early days of human civilization, arising from the need to transport and trade various materials. Initially, humans transported goods using their own bodies, later transitioning to animals, carts, boats, and other means until the creation of the first combustion-powered modes of transportation such as trains, cars, trucks, ships, and airplanes [2].

As technology and science advanced, transportation methods followed suit, becoming increasingly efficient to meet new logistical demands. We now live in an increasingly globalized world where distances are shortened, and events can be instantly known anywhere on the planet [3]. In this context, transportation logistics has become indispensable, allowing raw materials and consumer goods to travel great distances through different modes [4].

Throughout the 20th century, road transportation was widely recognized as the primary means of moving products and inputs in Brazil [5]. National mobility and production flow were almost entirely dependent on this system. Road freight transport is still one of the main modes used in logistics operations in Brazil today. It plays a fundamental role in the movement of a wide variety of products through highways. Among the most



transported products nowadays are agricultural derivatives such as soybeans, corn, wheat, sugar, live cargo, and fertilizers, as well as industrial products like food, steel products, machinery, and raw materials [6].

According to the National Logistics Plan for the year 2025, elaborated by the Empresa de Planejamento e Logística (EPL), despite the significant disadvantages of the road transport mode, such as the high consumption of time, energy, space, and financial resources, the economy still depends on this mode; as of 2015, highways were responsible for transporting at least 65% of all cargo in the country [1].

Among the transportation modes, this is also the segment that employs the most and has the largest share in the sector's wealth production, upon which numerous companies depend [7].

For most companies, transportation is the most significant factor in terms of total logistics costs, representing one- to two-thirds of these costs. Therefore, it is understandable that companies relying on road freight transport services seek ways to reduce these costs [8].

When analyzing the current scenario of road freight transport contracting, a recurring problem is identified that has negatively impacted the logistics operations of its users: the lack of structured methods for selecting and hiring carriers for each demand. This challenge is exacerbated by the existence of multiple criteria to consider, such as price, delivery time, service quality, safety, and sustainability. The lack of a systematic approach to deal with these criteria hinders decision-making, resulting in less accurate choices and higher costs. The hiring of road freight transport services is a reality in the Brazilian cargo transport scenario, as the country has an extensive road network that is predominantly used to transport products and inputs throughout the national territory. However, this mode faces significant challenges, such as poor infrastructure, high operational costs, road safety issues, and negative environmental impacts.

These obstacles directly impact the hiring of carriers, making the selection and decision-making process complex and challenging for users. Understanding this context is essential to develop strategies aimed at improving the efficiency and competitiveness of logistics operations in the road freight transport sector in Brazil.

There is an extensive body of literature that deals with the evaluation and selection of logistics operators through the MCDA methodology. Among the analyzed studies, the works of Zamcopé et al. [9], Cuba and Mazzuco [10], Costa [11], and Salazar and Pinheiro [12] stand out, along with others addressing the selection of road freight transport services, such as the studies by Sousa Júnior [13] and Florêncio [14]. However, despite the availability of materials on both topics, no relevant studies were found that apply the MCDA/MACBETH methodology to the selection of suppliers for the road transport mode, which represents the main contribution of this work.

According to a study by CNT [15], the road transport mode is the most widely used alternative in Brazil for cargo transportation, being responsible for about 65% of all products transported within the national territory. In this scenario, companies from various sectors of the economy depend on road freight services, which are often the only option to get products to consumers.

The proper selection of partners can play an important role in increasing the competitiveness of contracting companies [16]. Furthermore, this careful selection can also result in reducing the number of suppliers and establishing strong partnerships [17], bringing additional benefits.

It is natural for companies to seek to reduce their transportation costs, as these represent a significant portion of logistics costs, but this should not be the only aspect considered when contracting road freight transport. Since decision-making involves a wide range of criteria related to organizational strategies, it is crucial to develop methods that facilitate and support this complex process [18].

Seeking to gather the factors that should be taken into account by companies when hiring road freight transport services, this study aims to present a decision model capable of

supporting decision-making. It can be used by any company that relies on these contracts and wishes to make them more efficient, not just analyzing the costs involved.

The upcoming sections will include a review of the relevant literature, an explanation of the methodology employed, the presentation of the findings, and conclusions. Each part will delve into the existing literature, outline the study's approach, share research outcomes, and summarize key insights derived from the research.

## 2. Literature Review

The following topics will be addressed: logistics and transportation; hiring process in road freight transport; and freight value.

### 2.1. Logistics and Transportation

#### 2.1.1. Concept of Logistics and Supply Chain

Logistics encompasses a set of functional activities integral to the transformation of raw materials into finished products, thereby adding value for consumers. It is essential for companies as it optimizes time and place dimensions, ensuring products and services are available when and where customers intend to consume them [8]. Moreover, logistics is a company function that manages the physical flow of raw materials and product distribution to customers, interacting with various organizational areas [19]. According to the Council of Supply Chain Management Professionals (CSCMP), logistics involves strategic, operational, and tactical activities such as product transportation management, fleet management, storage, material handling, order tracking, logistics network development, inventory management, supply and demand planning, the management of logistics service providers, supplier development, production planning and scheduling, packaging, assembly, and customer service [20].

Chopra and Meindl broaden the concept by defining the supply chain as comprising all stages involved in fulfilling customer orders, encompassing manufacturers, suppliers, carriers, warehouses, retailers, and even the customers themselves. Within organizations, the supply chain integrates crucial functions such as new product development, marketing, operations, distribution, finance, and customer service. Given its pivotal role and comprehensive scope, logistics is fundamental to the operations of any company [21]. The subsequent discussion will delve deeper into transportation, emphasizing its critical dependence within organizational logistics.

#### 2.1.2. Cargo Transportation

Transportation plays a critical role in logistics by managing the flow of goods and enhancing product value [22]. This process is not only about moving goods but also about optimizing time, determining how quickly and reliably products move between locations [23]. For most businesses, transportation constitutes a significant portion of logistics costs, typically accounting for around 60% of expenses and sometimes exceeding operational profits [24]. Notably, in Brazil, logistics costs, encompassing transportation along with inventory, storage, and administrative services, consumed approximately 12.6% of the country's GDP in 2020 [25].

The nature of transported cargo, characterized by weight, volume, fragility, and other factors, dictates the choice of transportation mode [26]. The chosen mode aims to achieve a satisfactory level of service, combining various elements to ensure product availability. Moreover, transportation is closely intertwined with customer service, given the need to handle diverse cargo types, manage travel times, ensure punctuality, and mitigate risks like theft or damage [27].

Each transport mode possesses distinct characteristics that make it suitable for specific types of cargo and routes. The following subsection provides a concise overview of these modes and their primary attributes.

2.1.3. Main Freight Transportation Modes in Brazil

In Brazil, transportation relies on five main modes for cargo movement: air, waterway, pipeline, rail and road [28].

Air:

The role of air transportation in logistics has undergone many changes with globalization because production chains have branched out worldwide. Therefore, the supply of components and product distribution must be capable of achieving satisfactory levels of reliability in delivery times [29].

This type of transportation is primarily used for transporting high-value cargo (electronic items, watches, high-end fashion, etc.) as well as perishable items (flowers, premium fruits, medicines, etc.) [27].

Regarding dedicated cargo planes, the capacity of these planes is specifically designed for this type of transportation. For example, an MD-11 cargo plane can carry up to 92 tons of cargo, and the Boeing 747 can carry up to 112 tons. The Antonov 223, the world's largest cargo plane, has a capacity of up to 250 tons of cargo. These models also feature wide doors and access ramps for vehicles and containers [29].

One of the disadvantages of air cargo transportation is the high operational cost, which is the highest among all modes. Among the main costs are the aircraft itself, handling costs, cargo systems, fuel, labor, and maintenance [27].

Waterway:

Among the products transported by this mode, it is possible to highlight bulk liquid, chemicals, sand, coal, cereals, and high-value goods [27].

Among the waterway transportation means, container ships, roll-on/roll-off (Ro-Ro) ships for vehicles, bulk carriers, oil tankers, gas carriers, refrigerated ships, and livestock carriers can be mentioned.

Waterway transportation is characterized by the movement of cargo through seas (maritime), lakes (lacustrine), and rivers (fluvial). There is also coastal shipping, which specifically refers to the movement of cargo along coastal waters, commonly used for transportation between ports within the same country or for short-duration trips [30].

Waterway transportation is capable of carrying large volumes of cargo over long distances. However, the delivery time for products is longer due to the speed of ships and the unloading process at terminals, which tends to be slower than other modes. Among its advantages, it is possible to mention its low operational cost and low $CO^2$ emissions [23].

Pipeline:

Pipeline can be understood as the generic classification of a pathway composed of interconnected pipes, which is intended for the movement of cargo capable of meeting the specific requirements of this mode [31].

The use of pipeline transportation is still very limited and is primarily intended for the transport of liquids and gasses in large volumes, as well as materials that can be suspended, such as ores, crude oil, and derivatives [27].

In Brazil, pipeline transportation is mostly established and used by large companies in the petroleum and petrochemical industry, which use gas pipelines and oil/polymer pipelines. This is mainly because they control industrial and commercial processes at both ends of the mode, and they can engage in activities such as exploration, export, import, refining, and distribution [31].

Among its advantages, it is possible to mention that pipeline transportation is the safest of all, as it is protected from weather factors unlike other modes, and has very low labor costs. Among the disadvantages is the slow movement of products, making it impossible to transport perishable goods [27].

Rail:

In the Brazilian context, railway transportation is predominantly used for the movement of large volumes of standardized products, covering considerably long distances [27]. The authors emphasize that among these cargoes are coal, iron and manganese ores, petroleum derivatives, and grain cereals, which are transported in bulk.

Due to the absence of frequent congestion found on highways, railway transportation is suitable for products that do not require immediate delivery. However, it is important to note that railways operate at relatively low speeds and have limitations in terms of route flexibility [23].

Road:

It is the most significant mode in terms of cargo transportation in Brazil, capable of reaching almost all points of the national territory, thanks to investments in road pavement and the establishment of the automotive industry since the 1950s. It differs from rail transportation because it specializes in short-distance transportation of finished and semi-finished products. Road transportation usually has higher freight prices compared to rail and waterway modes, making it recommended for perishable goods but not suitable for bulk agricultural product transportation due to their low cost for this mode [27].

Road transportation is the most widely used mode in Brazil today, with its main characteristics including quick and door-to-door deliveries, extensive national coverage, and relatively low fixed costs. On the downside, its main disadvantages include a lack of security and high levels of $CO^2$ emissions in the environment [23].

It can be observed that each of the cargo transportation modes has its own characteristics, advantages, and disadvantages. Therefore, the choice of one mode over another is circumstantial and should take into account the factors most relevant to the shipper.

### 2.1.4. Preferences in the Brazilian Transportation Matrix

When it comes to freight transportation in Brazil, it is possible to observe a strong dominance of the road transport mode, followed by the railway, waterway (including coastal and inland navigation), pipeline, and, finally, air transport, as shown in Scheme 1, which compares the share of each mode in 2022.

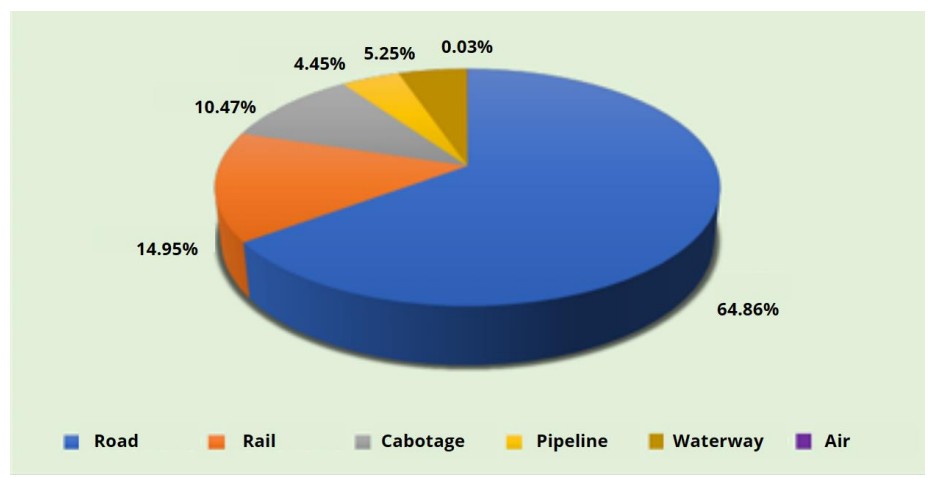

**Scheme 1.** Percentage comparison of the Brazilian transportation matrix in 2022. Source: [15].

Logistical factors such as the lack of investment in the railway network explain the preference for road transportation. Other factors also unduly favor this transportation option, including (i) the practice of overloading, which results in the destruction of the national road network and higher road maintenance costs; (ii) tax evasion, due to inefficiencies in controlling the issuance of transport documents or even invoices for products transported by independent drivers; (iii) the practice of the "carta-frete" (freight note); and (iv) the practice of charging below-cost freight rates, which hinders fleet renewal,

leading to an average vehicle age of 19 years and generating negative consequences for fuel consumption, pollution, and accidents [32].

Many of the distortions in the cargo transportation matrix and its respective inefficiencies can also be explained by the many years of state ownership of ports, railways, and pipelines in Brazil, as well as subsidies provided to the road sector in the past that persist to this day [33].

### 2.2. Logistics and Transportation

### 2.2.1. Logistics service Outsourcing

Outsourcing within logistics can be understood as the management process in which certain parts of the supply chain responsibility are transferred to third parties through established partnerships, enabling companies to focus solely on core business activities [34]. Its roots go back to the 1940s during World War II when the United States collaborated with European nations to combat Nazi forces, leading to its widespread adoption to boost wartime production capacity [35]. In Brazil, outsourcing gained traction with the arrival of multinational companies, notably in the automotive sector in the early 1980s, which streamlined operations by sourcing components externally to concentrate efforts on vehicle assembly [36]. This strategic shift toward outsourcing was driven by a desire for greater efficiency in logistics operations, a trend embraced by numerous companies and industries today, each with their unique motivations for entrusting operational, strategic, or management activities to logistics service providers (LSPs) [37].

Barros' research, which offers an overview of outsourcing in Brazil, highlights that 81% of respondents identified cost reduction as the primary motivation for outsourcing [37]. Additionally, Reis [38] delineates a range of advantages associated with outsourcing, including the ability to focus on core business functions, converting fixed costs to variable costs, enhancing flexibility, efficiency, and productivity in logistics processes, accessing cutting-edge technology with regular updates, minimizing investment in fixed assets, expanding geographic coverage, penetrating new or untapped markets, improving customer service, and mitigating labor-related issues.

It is essential to underscore that selecting the wrong outsourcing partner—one who fails to fulfill their responsibilities—can severely damage the hiring company's reputation and result in additional costs to rectify affected processes [39]. This caution highlights the critical importance of due diligence in choosing reliable and competent partners for outsourcing engagements.

### 2.2.2. Outsourcing of Road Cargo Transportation

The outsourcing of transportation by companies involves several critical issues [40], including controlling the risk of dependence on shippers, ensuring operational flexibility, avoiding complexity in LSP management, promoting attractiveness to carriers, maximizing transportation asset utilization, and leveraging carriers' geographical and competency specialization. The selection of LSPs typically begins with defining and evaluating elimination criteria to develop a comparative overview of potential providers, particularly those offering road transportation services. Companies that meet initial requirements undergo more detailed analysis to identify the most beneficial proposal, while others are promptly rejected [29].

Among the reasons for the high level of outsourcing in cargo transportation to logistics service providers (LSPs), approximately 80% of companies opt for this service, primarily utilizing road networks due to the prevailing state of the country's transportation infrastructure [40]. This trend can be attributed to the abundant supply of road transport and the generally lower prices in the market, which often fail to cover carriers' actual costs [41].

Considering the importance of selecting an LSP for road transportation operations, various factors come into play, such as the compatibility of information systems, market reputation, financial stability, sector experience, cultural alignment, communication efficiency, geographic coverage, and competitive pricing [29]. Galo's study, which evaluated selection

criteria based on multiple authors' perspectives, identified critical factors including unit freight cost, on-time delivery performance, lead time, damage rates, flexibility, emergency response, service levels, communication effectiveness, accurate quantity delivery, electronic data interchange, complaint handling, delivery reliability, damage-free delivery, customer satisfaction, and the resolution of complaints [42].

*2.3. Freight Cost*

One of the most critical considerations in hiring road transportation services is the freight cost, defined as the price paid for transporting goods and influenced by multiple factors [26]. In the Brazilian market, despite road transportation being prevalent for cargo transport, freight prices often fall below recommended economic levels due to services outsourced by autonomous drivers and transport companies aiming to cut costs [34]. A CNT survey highlights challenges in the open market with low entry barriers and fluctuating supply and demand, contributing to freight rate competitiveness below international standards [32]. Large companies in Brazil have significant bargaining power over smaller carriers due to market competition, resulting in strategic advantages during negotiations [43].

Novaes [29] criticizes the abundance of inexperienced operators offering services at very low rates in Brazil's road freight sector, impacting both autonomous and commissioned drivers. Research by Araújo, Bandeira, and Campos shows significant discrepancies between actual freight costs and what should be charged, affecting drivers' earnings and company profitability [32]. Seeking to address these challenges, the federal government introduced the National Policy for Minimum Freight Rates for Road Cargo Transportation (PNPM-TRC) to ensure fair compensation for transportation services [44].

Understanding freight pricing involves considering various factors such as distance, cargo specificity, delivery time, operational costs, and market dynamics. Transport companies calculate freight costs based on cargo value, difficulty, fixed costs (e.g., salaries, insurance), and variable costs (e.g., fuel, maintenance) [26,45]. Additionally, standardized fees like weight freight (based on cargo weight or volume), ad valorem freight (based on cargo value), and risk management and security fees contribute to freight composition [46]. Despite the focus on cost reduction, logistics practices must also prioritize service quality and reliability to meet customer expectations [29].

In conclusion, while the literature extensively covers logistics, transportation, and freight management, gaps exist in understanding logistics outsourcing and road freight hiring processes, decision-making criteria, and cost optimization strategies using innovative methods. Pricing road freight involves multiple factors, with cost being paramount for consumers; however, quality and reliability remain essential considerations in modern logistics [29].

**3. Methodology**

This article made use of theoretical data, analytical methods, comparison, and synthesis to establish a robust framework. Theoretical data come from existing theories, models, and conceptions. They were utilized to underpin established principles such as MCDA and relevant concepts within Brazil's current RFT market. The analysis method employs a systematic approach to comprehending information, breaking down complex elements, evaluating their relationships, and drawing conclusions, which was instrumental in identifying critical factors in the hiring process. Comparative analysis enhances understanding by examining relationships and differences among elements, and emphasizing distinctions and patterns, which supported the analysis of results. Synthesis involves merging diverse elements to create a unified understanding, integrating information into a cohesive model capable of supporting the logistics operator hiring process efficiently and concisely [47].

The chosen methodology for determining the decision model was multicriteria decision analysis (MCDA), which is part of the scope of Operations Research (OR). The origins of the first version of Operations Research can be traced back to the early days of World

War II when military operations required a more efficient allocation of resources than was practiced at the time [48]. Operations Research can also be defined as an applied decision theory that uses scientific, mathematical, or logical means to structure and solve decision problems [49].

MCDA did not emerge alongside OR but rather stemmed from the studies of Roy [50], Keeney and Raiffa [51], and Saaty [52], who are considered pioneers of the methodology. Original OR was based solely on searching for a single objective function subject to a set of constraints, reduced to an evaluating function [53]. MCDA, on the other hand, emerged in the scientific community as an alternative solution for complex and ill-structured problems that original OR could not solve [54].

MCDA consists of three basic phases: structuring, evaluation, and recommendations. The methodology can be used for both ex post and ex ante analysis, meaning that it can analyze both processes preceding decision-making and decisions that have already been made to ensure that their objectives were achieved [53].

The MCDA proposed in this study will consist of seven phases, as presented by Ensslin [54]. These phases are as follows:

- Label Definition: In this phase, the objectives and criteria that will guide the decision-making process will be established, defining clearly what is intended to be achieved in the RFT (road freight transport) contracting.
- Actor Identification: The parties involved in the decision-making process, such as those responsible for contracting, RFT service providers, and other relevant stakeholders, will be identified.
- Identification of Evaluation Elements: Key elements to be evaluated in the decision-making process will be identified, considering aspects such as service quality, reliability, costs, and other relevant factors.
- Descriptor Construction: Clear and measurable descriptions will be developed for each identified evaluation element to facilitate comparison and subsequent analysis.
- Determination of Value Functions: Value functions for each criterion will be established, reflecting the preferences and relative importance attributed to each of them.
- Definition of Replacement Rates: In this phase, replacement rates between different criteria will be established, considering the possibility of trade-offs between them.
- Construction of the Value Tree: Finally, a hierarchical value tree will be developed, integrating the criteria and the relationships of dependence between them, providing a coherent structure for decision-making.

To create an efficient decision model to assist the RFT contracting process, this study will adopt the multicriteria decision analysis (MCDA) approach, following the single-criterion school of synthesis for the development of evaluation axes; the MACBETH methodology for defining value functions; and the swing weights method for defining replacement rates.

Numerous methodologies and tools are currently available for modern multicriteria decision analysis (MCDA), including the Analytic Network Process (ANP), Technique for Order of Preference by Similarity to Ideal Solution (TOPSIS), Elimination and Choice Expressing Reality (ELECTRE), Preference Ranking Organization Method for Enrichment Evaluations (PROMETHEE), Multicriteria Optimization and Compromise Solution (V.I.K.O.R.) [55], and, more recently, Artificial Intelligence (AI) techniques. Among these, MACBETH uniquely addresses how adjustments in criteria weights can influence the overall decision-making process. Concurrently, the swing weights method evaluates the importance of criteria by examining the impacts of shifting from the least favorable to the most favorable outcomes for each criterion. This capability is particularly advantageous in RFT processes, where certain criteria may disproportionately influence the final decision based on their perceived importance to the stakeholders.

By following these methods and steps, it is expected that the resulting decision model will be capable of providing a comprehensive and weighted assessment of RFT providers, considering the defined criteria and enabling the selection of the best contracting alternative.

### 3.1. Label Definition

Taking into consideration that the main objective of this work is to construct a decision model to support the hiring of road freight transportation services, the label defined for the model was the hiring of road freight transportation services.

### 3.2. Identification of Actors

The agents who directly or indirectly participate in the decision-making process and assist in building the model with their suggestions can be defined as acted upon and interveners. the interveners are subdivided into two categories: decision-makers and facilitators [54].

- Acted Upon: the companies using freight transportation;
- Decision-Makers: experts in road freight transportation and managers of companies in the field;
- Facilitators: the authors of this work. In 2022, three decision-makers, who, according to Ensslin et al. (2001) [54], can be considered rational sources, were invited to provide opinions as professionals in the road freight transportation industry and were of great importance in the development of the model: they were three transportation directors from anonymous companies. All companies involved have more than 1000 trucks delivered per year and are heavily dependent on road transportation for their core businesses.

### 3.3. Identification of Evaluation Elements

There is a sequence of activities to identify evaluation elements. These activities include (i) the identification of the primary evaluation elements (PEEs); (ii) the construction of cognitive maps; and (iii) the identification of the fundamental points of view (FPVs) [54].

PEEs encompass objectives, goals, and decision-makers' values, as well as actions, options, and alternatives. They serve as the initial step to create a mental map, defined as a hierarchy of concepts connected by links of influence between means and ends [54].

This information is obtained through a brainstorming process with decision-makers and must follow certain requirements [56]:

- Express all PEEs that come to mind.
- Quantity is desired, so the more PEEs, the better; avoid criticizing ideas.
- Ideas that have been presented can be improved and combined.

After defining the PEEs and constructing the cognitive map, the next step is to determine the FPVs, which express the values considered relevant by decision-makers in that specific context, and establish the characteristics of actions that interest these decision-makers [57].

FPVs constitute the evaluation axes of the problem [58] and must meet requirements such as being essential, controllable, complete, measurable, non-redundant, concise, understandable, isolatable, and operational [51,58].

Due to the complexity of the subject, it was necessary to break down the FPVs into elementary points of view (EPVs), which allow for a better evaluation of the performance of potential actions from the viewpoint considered, improving the understanding of what a fundamental viewpoint intends to consider. They should be used whenever one wants to decompose the evaluation axis. Here are the evaluation elements selected after defining the FPVs and EPVs [57].

FPV 1—Total Freight Cost:
FPV 2—Inherent transportation aspects:

EPV 2.1—Delivery time;
EPV 2.2—Fleet age;
EPV 2.3—Cargo location service.

FPV 3—Customer service channels:

EPV 3.1—Email support;
EPV 3.2—Phone support;
EPV 3.3—WhatsApp support.

FPV 4—Company reputation;
FPV 5—Time in the market;
FPV 6—Sustainability.

### 3.4. Determination of the Descriptors

An evaluation instrument should be defined containing two tools for each evaluation axis, which extends from the FPVs to the EPVs. These tools are a descriptor, which will be discussed in this subsection; and a value function, which is covered in the following subsection [54].

A descriptor can be understood as a set of impact levels (ILs) intended to describe plausible performances of the criteria. Since in this work, almost all the FPVs (criteria) were broken down into EPVs (sub-criteria), descriptors were defined only for these.

Five impact levels were defined for each descriptor, ordered in descending order, from the most attractive (best possible performance action) to the least attractive (worst possible performance action). For the elements present in the evaluation model, a single descriptor was defined based on the Likert scale, as follows:

- N5—Excellent;
- N4—Good;
- N3—Average;
- N2—Poor;
- N1—Very Poor.

### 3.5. Determination of Value Actions

A value function can be understood as a tool accepted by decision-makers to help articulate their preferences [51]. It is used to rank the preference intensity (difference in attractiveness) between pairs of impact levels or potential actions. It should be constructed for a decision-maker or group of decision-makers with the aim of evaluating actions from a specific point of view [54].

Value functions (local attractiveness rates or local evaluations) can be described as mathematical representations, made through graphs or numerical scales, of decision-makers' value judgments about one or more criteria. They numerically measure, from a fundamental point of view, the degree of attractiveness of each impact level related to a scale with predefined levels that correspond to the decision-maker's value system [58].

Among the multiple methods available for determining value functions, the semantic judgment method was chosen for this work, which is considered appropriate to assist the decision-maker in articulating their preferences during the evaluation from a specific point of view [58].

In semantic judgment methods, a value function is constructed based on pairwise comparisons of the attractiveness differences between potential actions. For this purpose, the decision-maker is asked to qualitatively express the preference intensity of one action over another [58].

The semantic judgment method used was the Measuring Attractiveness by a Categorical-Based Evaluation Technique (MACBETH), created by Bana e Costa and Vansnick [57]. MACBETH uses semantic judgments of decision-makers and verbally translates the attractiveness difference between two potential options a and b (a more attractive than b). For this work, one of the ordinal scales available in the software will be used, which can be observed in Table 1:

**Table 1.** Semantic ordinal scale used by MACBETH.

| Description | Scale |
|---|---|
| Definition of extreme attractiveness | Extreme |
| Very strong attractiveness difference | Very strong |
| Strong attractiveness difference | Strong |
| Moderate attractiveness difference | Moderate |
| Weak attractiveness difference | Weak |
| Very weak attractiveness difference | Very Weak |
| No attractiveness difference | Null |

Source: MACBETH.

Based on semantic categories, a matrix, known as a semantic matrix, is then constructed, containing the attractiveness differences indicated by decision-makers in relation to the impact levels of the same descriptor. This matrix serves as input for the calculation of value functions by MACBETH through linear programming. From the moment a value function is associated with an FPV, it is referred to as a criterion, while its EPVs are called sub-criteria [53].

*3.6. Definition of Replacement Rates*

Replacement rates, compensation rates, or simply weights refer to the loss of performance that one criterion or sub-criterion must have to compensate for the gain in another, so that its overall value remains unchanged [59].

In this study, among the many methods available for defining replacement rates, the method of balanced weights (swing weights) was adopted, which involves ranking the criteria (or sub-criteria) in order of preference, assigning them weights on a decreasing scale in a two-step constant construct, as demonstrated below.

Step 1: Ranking the criteria based on priority determined by the decision-makers, assigning a score of 100 to the most important criterion, and scoring the other criteria according to their priority in relation to the most important criterion, as shown in Table 2. The scores were obtained based on an arithmetic average of the ratings given by the 3 decision-makers.

**Table 2.** FPV scores.

| FPV | Criteria | Score |
|---|---|---|
| 1 | Total freight cost | 100 |
| 2 | Inherent aspects of transportation | 40.5 |
| 3 | Customer service channels | 13.5 |
| 4 | Company reputation | 9 |
| 5 | Time in the market | 9 |
| 6 | Sustainability | 9 |
| | Total | 181 |

Step 2:C of the replacement rates, based on the determination of the percentage contribution of each criterion, calculated according to the procedure presented in Table 3.

**Table 3.** Calculation of the replacement rates of the FPVs.

| FPV | Criteria | Calculation of the Percentual Value | Replacement Rates |
| --- | --- | --- | --- |
| 1 | Total freight cost | 100/181 × 100 = 55.24% | 55% |
| 2 | Inherent aspects of transportation | 40.5/181 × 100 = 22.37% | 22.50% |
| 3 | Customer service channels | 13.5/181 × 100 = 7.45% | 7.50% |
| 4 | Company reputation | 9/181 × 100 = 4.97% | 5% |
| 5 | Time in the market | 9/181 × 100 = 4.97% | 5% |
| 6 | Sustainability | 9/181 × 100 = 4.97% | 5% |

*3.7. Construction of the Value Tree*

Once the evaluation criteria of the model were established, the value tree was developed, representing a diagram of the defined structure. This structure consists of the criteria, sub-criteria, and their respective weights, as illustrated in Figure 1.

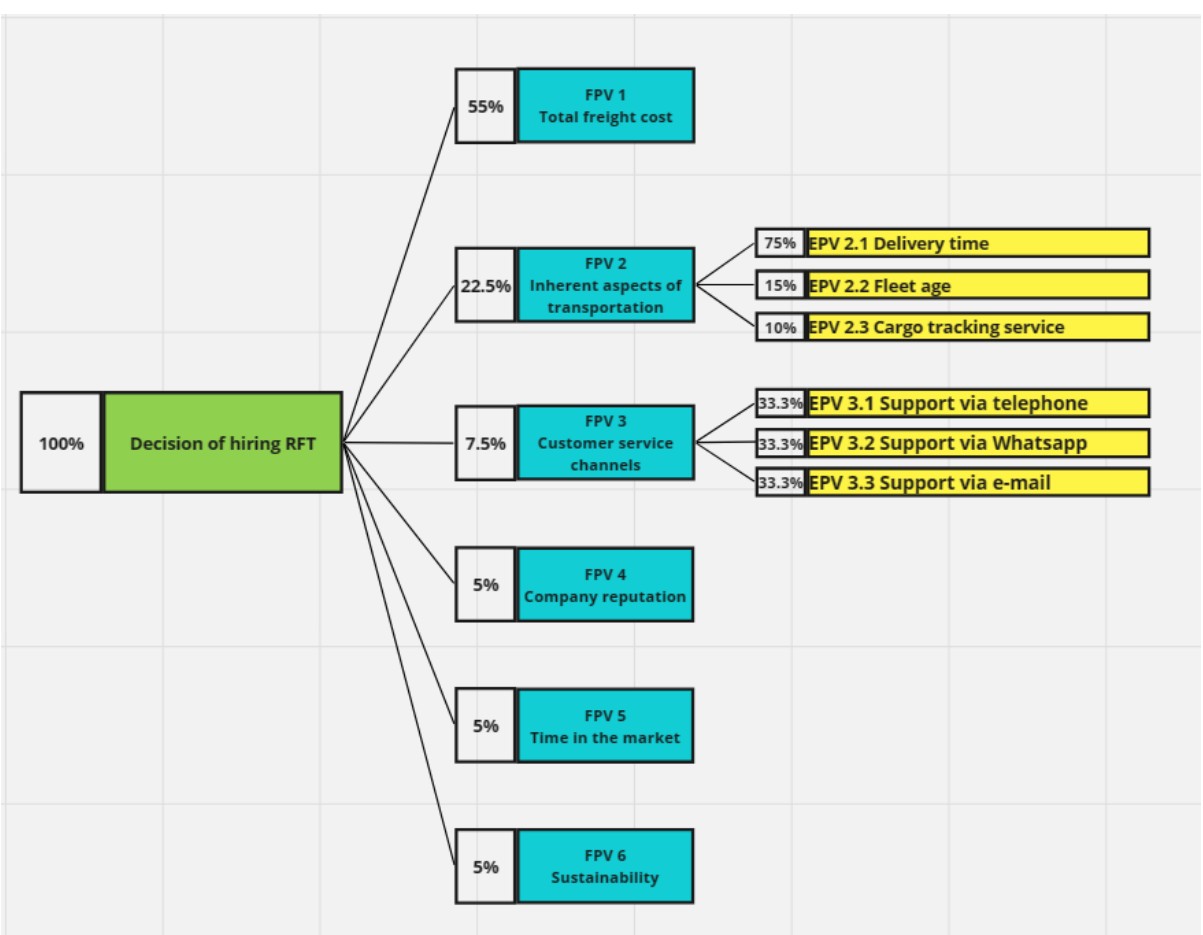

**Figure 1.** Value tree.

After completing the presentation of the evaluation model, the procedures for conducting the sensitivity analysis evaluations are described below.

*3.8. Procedure for the Calculation of Assessments*

Considering that the ultimate objective of this study is to structure a model that assists in the hiring of road freight transport (RFT), the following procedures are specified for transforming qualitative data into an overall quantitative assessment that numerically

expresses, on a scale from zero to ten, the degree of quality of the road freight transportation service to be contracted:

(a)    Equation for calculating the assessments of the criteria (FPVs)

$$\text{AFPV} = \sum_{i=1}^{m} \sum_{j=1}^{n} pi.[(\text{FViRj})1/n]1/10 \tag{1}$$

- AFPV = assessment of the criteria (FPVs);
- FViRj = value function of respondent j impacted on sub-criterion i;
- pi = sub-criterion i replacement rate;
- n = number of survey respondents;
- m = number of criteria in the model.
- The sum of the replacement rates must be equal to 1 (p1 + p2 + … + p6 = 1);
- The value of the replacement rates must be greater than zero and less than 1 (1 > pi > 0 for i = the number of sub-criteria of the FPV).

(b)    Equation for calculating the global scores:

$$AG = \sum_{i=1}^{n} pixi(FPV) \tag{2}$$

- AG= global scores;
- xi (FPV) = evaluation of criteria (1, 2, 3, 4, 5, and 6);
- pi = replacement rate (weight) of criteria (1, 2, 3, 4, 5, and 6);
- n = 6 (number of criteria in the model).
- The sum of the replacement rates must be equal to 1 (p1 + p2 + … + p6 = 1);
- The value of the replacement rates must be greater than zero and less than 1 (1 > pi > 0 for i = 1, 2, … 6).

By using these formulas and processing the data tabulated by the Hiview3 3.2 software, it will be possible to calculate the assessments of the criteria (FPVs) and the overall evaluation (global score), which will ultimately serve as an indicator of which company should be hired.

### 3.9. Sensitivity Analysis

Despite joint efforts in building the decision model, it remains to be seen whether the criteria contained in it are sufficient to provide the necessary robustness for practical use in the hiring of RFT. However, this can only be assessed through a sensitivity analysis, which will be carried out after the field research and the proper treatment of the collected data.

A sensitivity analysis allows us to verify if a slight change in the replacement rates has an impact on the overall result [60]. Conducting a sensitivity analysis of the model requires varying the replacement rate of each criterion to assess the impact on the evaluations of potential actions [54].

The model's effectiveness is validated when minor changes in the replacement rates of the criteria do not overly influence the overall evaluation. On the other hand, a significant impact resulting from the replacement indicates the need for adjustments [54,60].

When altering the weight of one criterion, the replacement rates of the other criteria in the model also need to be adjusted so that the total remains at a value of 100%. Under this condition, any necessary adjustments should follow the following formula:

$$pn' = (pn(1 - pi'))/(1 - pi) \tag{3}$$

- pi = replacement rate (weight) of criteria (1, 2, 3, 4, 5, and 6);
- pi′ = modified replacement rate (weight) of criterion i;
- pn = original replacement rate (weight) of criterion n;
- pn′ = modified replacement rate (weight) of criterion n.

The sensitivity analysis of the decision model will be conducted by varying the replacement rates of the criteria (FPV) by ±10%, altering the parameters of these rates, and assessing the impact on the overall evaluation [54]. The sensitivity analysis of the constructed model will be demonstrated in Section 4.2.

## 4. Results

The field research was conducted between December 2022 and April 2023, through emails, phone calls, and websites. Through these channels, it was possible to simulate the process of hiring transportation services with eight companies that provide RFT services.

In response to the preferences of our field research participants, the companies involved in the simulation were anonymized and assigned letters A to H. To ensure uniformity across all selected companies, specific criteria were applied: each company offered cargo insurance in case of damage, possessed a minimum of one year's experience in the RFT market, maintained active customer service via phone or internet, and had publicly available online reviews assessing their performance.

The simulation involved sending a request for a quote for the transportation of three tons of soybeans, covering a distance of 605 km along the São Paulo (SP) to Belo Horizonte (MG) route.

For a better visualization of the route in question, Figure 2 presents the complete map of the journey.

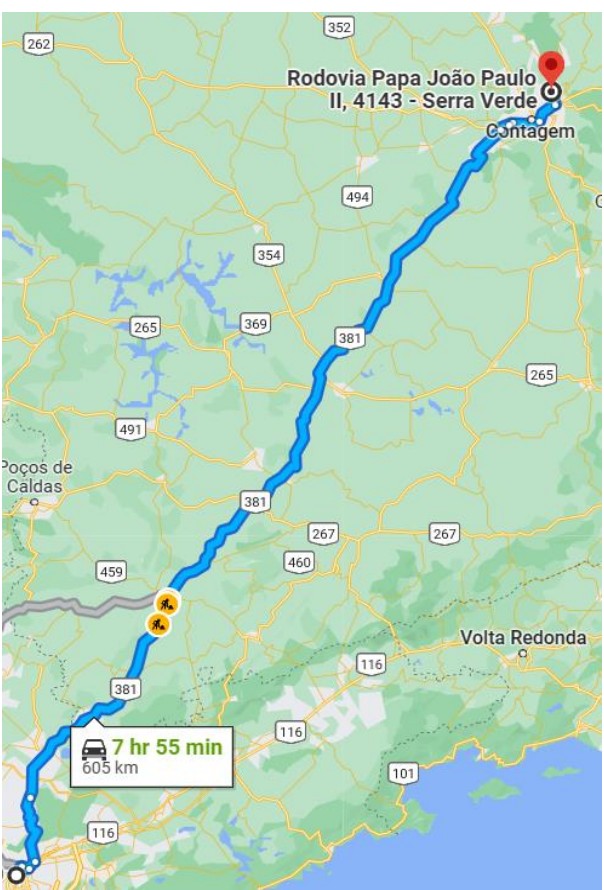

**Figure 2.** Simulated route—São Paulo (SP) to Belo Horizonte (MG). Source: Google Maps.

Based on the data collected through the questionnaires and the analyses conducted, the field research contributed to identifying the best company options for transporting this cargo, aiming for an informed and effective decision-making process.

After defining the decision model and collecting the field research data, the data were tabulated and processed, assessments of the criteria and the overall evaluation for each

company were calculated, and a sensitivity analysis was conducted. This section will include the following subsections: data analysis and application of sensitivity analysis.

*4.1. Data Analysis*

The data collected during the field research were processed using the Hiview3 software following the decision model. After performing the necessary calculations, the results of the assessments for criteria, sub-criteria, and the overall evaluation were compiled for each of the eight companies.

The results analysis will utilize the tabulation of the scores for each of the FPVs and, in the end, will present the company with the highest global score for providing this service.

4.1.1. Analysis of FPV 1—Total Freight Cost

Managers responsible for hiring land transportation services must be constantly vigilant regarding the efficiency of internal company processes, as well as market parameters. Therefore, it is essential to conduct a thorough analysis of the freight costs applied to the various routes used [43].

As explained in the theoretical framework, transportation costs represent a significant portion of logistics costs. Therefore, reducing these costs becomes a priority for many companies, which requires the FPVs to have a greater impact on the decision model, with a weight of 55%.

The desired minimum price for this simulation was obtained through the ANTT (Agência Nacional de Transportes Terrestres) freight table, which, based on Ordinance No. 5 of 17 February 2023, for bulk solids [61], provided a price of USD 420.40 for this transport, considering displacement costs in USD/km (USD 353.98 for 605 km) and loading and unloading of soybeans (USD 66.42) in a two-axle truck. The original price was provided in the official Brazilian coin, Real (BRL), and converted at a rate of BRL5.09/USD.

It was then decided that to assign scores according to the Likert scale, it would be necessary to define price ranges, which, for simulation purposes, were established as follows:

- Excellent: ≤USD 451;
- Good: ≤USD 589;
- Regular: ≤USD 687;
- Poor: ≤USD 785;
- Very Poor: >USD 982.

After collecting the prices from each of the companies for this route, the Hiview3 software was used to generate scores between 0 and 10 for each of them, which can be observed in Table 4.

**Table 4.** Scores on FPV 1—Total freight cost.

| Company | Scores on FPV 1 |
|---------|-----------------|
| Company A | 2.5 |
| Company B | 0 |
| Company C | 7.5 |
| Company D | 7.5 |
| Company E | 7.5 |
| Company F | 0 |
| Company G | 7.5 |
| Company H | 7.5 |

Although none of the companies was able to achieve the maximum score in this FPV, the highest score observed was 7.5, obtained by companies C, D, E, G, and H. This translates to the fact that most of them offered total prices in the "Good" range for this service. Company A, with the second lowest score of 2.5, offered a price range between

USD 687 and USD 785. Companies B and F, holding the minimum score in this FPV, offered prices exceeding USD 982.

It is worth noting that while this is the most significant FPV in the model, due to the similar prices, it remains to be seen how each of the companies performed in the other assessed aspects to determine the most suitable company for this transportation.

4.1.2. Analysis of FPV 2—Inherent Aspects of Transportation

In general, aspects inherent to cargo transportation should be taken into consideration when planning and executing a company's logistics to ensure efficiency and process safety, while minimizing risks and costs.

The FPV for aspects inherent to transportation is the second in terms of weight, accounting for 20% of the model and encompassing three EPVs, with their respective weights: delivery time (75%), fleet age (15%), and cargo location service (10%).

Delivery time (75%): Ballou (2011) [8] emphasizes that meeting delivery deadlines is one of the most crucial measures of logistic efficiency since it directly impacts customer satisfaction and a company's revenue. He argues that on-time delivery is a key factor in increasing customer loyalty and enhancing brand reputation.

The ideal delivery time for this simulation took into account two factors:

- The general rule for driver working hours states that the working day should not exceed 8 h according to art. 58 of Decree-Law No. 5452/1943—Consolidation of Labor Laws—CLT [62].
- According to the Brazilian Traffic Code (CTB), the maximum speed allowed for trucks is 80 km/h on single-lane highways and 90 km/h on highways of two or more lanes [63].

Respecting the mentioned regulations and establishing an average speed of 85 km/h for truck drivers, the minimum delivery time for this cargo was set as one business day, with one travel day corresponding to the maximum score on this EPA. Each additional day in the deadline reduces the score by exactly one rating on the Likert scale, as follows:

- Excellent = one business day;
- Good = two business days;
- Regular = three business days;
- Poor = four business days;
- Very Poor $\geq$ five business days.

Table 5 summarizes the scores obtained by each of the companies for EPV 2.1—Delivery Time.

**Table 5.** Scores on EPV 2.1—Delivery time.

| Company | Scores on EPV 2.1 |
|---|---|
| Company A | 10 |
| Company B | 7.5 |
| Company C | 2.5 |
| Company D | 2.5 |
| Company E | 0 |
| Company F | 0 |
| Company G | 2.5 |
| Company H | 2.5 |

As it is possible to observe, company A achieved a result superior to the others, committing to completing the service in just one business day and receiving the maximum score for this criterion. The second-highest rated company was company B, with a score of 7.5, equivalent to a two-day business delivery. Companies C, D, G, and H received a score of 2.5, as they would need up to four business days to complete this service, while

companies E and F received the minimum score, as they required at least five business days for delivery.

Fleet age (15%): Older fleets generate more negative externalities for society, such as increased air pollution, higher operational costs, and a greater likelihood of accidents. Owner-operator trucks have an average age of 23 years, in contrast to company-owned trucks, which have an average age of 13 years, and cooperative-owned trucks, which have an average age of 15 years [64].

Based on the premise that the newer the fleet, the lower the chances of accidents during transit, the highest scores for this EPV were given to companies with more modern trucks. However, there should be some tolerance since even companies which typically have better conditions to renew their fleets frequently, have an average vehicle age of over a decade. Here are the criteria used to evaluate each company:

- Excellent: ≤6 years;
- Good: ≤9 years;
- Regular: ≤12 years;
- Poor: ≤15 years;
- Very Poor: >18 years.

Table 6 summarizes the scores obtained by each of the companies for EPA 2.2—Fleet age:

**Table 6.** Scores on EPV 2.2—Fleet age.

| Company | Scores on EPV 2.2 |
|---------|-------------------|
| Company A | 10 |
| Company B | 7.5 |
| Company C | 7.5 |
| Company D | 5 |
| Company E | 7.5 |
| Company F | 10 |
| Company G | 2.5 |
| Company H | 10 |

Companies A, F, and H obtained the highest scores, reaching the maximum score as they were the only ones reporting fleet average ages of up to 6 years. Companies B, C, and E received a score of 7.5 as they reported fleet average ages of up to 9 years. Company D was evaluated with a score of 5, being the only one with a fleet average age of up to 12 years. Finally, company G received the lowest score in this EPV due to the average age of its fleet, which reached 15 years.

Cargo location service (10%): Some of the benefits of tracking systems include (1) the detection of violations, scheduling discrepancies, and unauthorized stops; (2) increased customer satisfaction by knowing their shipment can be located at any time; (3) more accurate delivery scheduling, providing customers with the opportunity for savings, efficiency, and fewer production disruptions; and (4) monitoring breakdowns, equipment failures, operator negligence, and accidents [65].

This EPV was evaluated in a binary manner, where companies received the maximum score (10) when they had a cargo location system that could be accessed by the customer, or, in the absence of such systems, they received the minimum score (0). Other FPVs and EPVs that will be discussed later followed the same logic.

It is worth noting that this type of service is provided by all contacted companies. It was observed that, in all cases, these services could be accessed through the companies' own websites.

Having presented the scores of the EPVs that make up the aspects inherent to transportation, it remains to examine the performance of each company in the FPV itself. Table 8 presents the compiled scores.

Table 7 summarizes the scores obtained by each of the companies for EPV 2.3—Cargo location service:

**Table 7.** Scores on EPV 2.3—Cargo location service.

| Company | Scores on EPV 2.3 |
|---|---|
| Company A | 10 |
| Company B | 10 |
| Company C | 10 |
| Company D | 10 |
| Company E | 10 |
| Company F | 10 |
| Company G | 10 |
| Company H | 10 |

**Table 8.** Scores on FPV 2—Inherent aspects of transportation.

| Company | Scores on EPV 2.1 | Scores on EPV 2.2 | Scores on EPV 2.3 | Scores on FPV 2 |
|---|---|---|---|---|
| Company A | 10 | 10 | 10 | 10 |
| Company B | 7.5 | 7.5 | 10 | 7.75 |
| Company C | 2.5 | 7.5 | 10 | 4 |
| Company D | 2.5 | 5 | 10 | 3.625 |
| Company E | 0 | 7.5 | 10 | 2.125 |
| Company F | 0 | 10 | 10 | 2.5 |
| Company G | 2.5 | 2.5 | 10 | 3.25 |
| Company H | 2.5 | 10 | 10 | 4.375 |

It can be concluded that the companies that achieved the best results in this FPV were companies A and B, with scores of 10 and 7.75, respectively, while company F received the lowest score among all those evaluated, 2.5.

### 4.1.3. Analysis of FPV 3—Customer Service Channels

The compatibility of service channels among companies that provide services to other businesses is crucial [29]. Hiring companies expect to receive quality service and often need quick responses to their demands. In this regard, the compatibility of service channels (such as email, phone, and chat) between companies can ensure smooth and effective communication, resulting in greater customer satisfaction.

Considering this aspect, the value of service channels' participation in the decision-making process corresponds to 7.5% and is composed of three means frequently used by companies and consumers. Each of these channels was evaluated in a binary manner (0 and 10) and carries an equal weight of 1/3 in the calculation of the value: the WhatsApp messaging app, phone (mobile or landline), and email.

Table 9 gathers the scores of each company for each EPV, as well as the score of FPV 3—Service channels:

**Table 9.** Scores on FPV 3 and respective EPVs—Customer service channels.

| Companies | Scores on EPV 3.1 (W) | Scores on EPV 3.2 (T) | Scores on EPV 3.3 (E) | Scores on FPV 3 |
|---|---|---|---|---|
| Company A | 0 | 10 | 10 | 6.666 |
| Company B | 0 | 10 | 10 | 6.666 |
| Company C | 10 | 10 | 10 | 9.999 |
| Company D | 0 | 10 | 10 | 6.666 |
| Company E | 10 | 10 | 10 | 9.999 |
| Company F | 0 | 10 | 10 | 6.666 |
| Company G | 10 | 10 | 10 | 9.999 |
| Company H | 10 | 10 | 10 | 9.999 |

All evaluated companies received scores equal to or greater than 6.66 in the FPV, indicating that they all stated that they provide customer service through at least two of the three analyzed channels. It is worth noting that companies C, E, G, and H were the ones that received the highest scores, with the maximum score of 9.99. It should be emphasized that, due to the division of the FPV into three parts, the authors chose to maintain result precision rather than rounding to 1.

4.1.4. Analysis of FPV 4—Company Reputation

Reputation is a social construct that applies to both individuals and companies. In general, reputation refers to the opinion or evaluation that people have of someone or something.

Companies are required by the current market context to establish a positive image in front of their stakeholders and continuously improve it. This image is essential for corporate reputation, which is formed by a set of organizational attributes developed over time that influence stakeholders' perception of the company [66].

Corporate reputation results from various different factors, including performance, behavior, communication, and organizational characteristics. It reflects the accumulated perception of the public regarding the company's past actions, its current behavior, and its future prospects for fulfilling its commitments [67].

Currently, it is common for consumers to use the internet to check the reputation of services or companies based on online reviews on specialized websites, search engines, and social networks.

Online reviews play an important role in shaping a company's reputation and are considered a valuable source of information for consumers [68]. In fact, research found that 87% of consumers consult online reviews before making a decision to purchase a product or service, and 93% of them claim that these reviews influence their final purchasing decision. Additionally, research also showed that 79% of consumers trust online reviews in the same way they trust personal recommendations.

Considering the importance of online reviews in building a company's reputation, or corporate reputation, this FPV accounts for 5% of the model. Three popular websites were chosen to collect user feedback: Reclame Aqui (popular Brazilian website that acts as an independent communication channel between consumers and companies, presenting customer reviews and scores), Google, and Facebook. Reviews are typically rated on a scale of 1 to 5, allowing for the distribution of scores as follows:

- Excellent: $\geq 4.5$;
- Good: $\geq 4$;
- Average: $\geq 3$;
- Poor: $\geq 2$;
- Very Poor: $< 2$.

In cases where companies received different ratings on review platforms, an arithmetic average was calculated for all the collected reviews. The formula used was as follows:

$$((FN) + (GN) + (AR))/N \tag{4}$$

- FN = Facebook rating;
- GN = Google rating;
- AR = Reclame Aqui rating;
- N = Number of platforms where the company was rated.

Table 10 gathers the ratings for each company for FPV 4—Company reputation.

**Table 10.** Scores on FPV 4—Company reputation.

| Company | Scores on FPV 4 |
|---|---|
| Company A | 10 |
| Company B | 10 |
| Company C | 10 |
| Company D | 5 |
| Company E | 7.5 |
| Company F | 5 |
| Company G | 10 |
| Company H | 2.5 |

In the evaluation of this FPV, companies A, B, and C obtained the highest ratings, being the only participants to achieve online ratings of 4.5 or higher. In contrast, company H received the lowest rating, with an online evaluation below 3. Company D received ratings between 5 and 7.5, resulting from online ratings between 3.0 and 4.4.

4.1.5. Analysis of FPV 5—Time in the Market

It is recommended that any type of outsourcing should take into consideration the hiring of companies with at least two years of experience [29]. Some of the problems that may arise from hiring a partner with fewer years of experience include:

- Inexperience: insufficient knowledge to deal with the challenges that arise in the day-to-day business, leading to errors and delays that can harm the contracting company.
- Lack of references: with only a few years of operation, it can be difficult for the contracting company to verify the quality of the services provided and the reputation of the hired company. This increases the risk of hiring a company that does not deliver what was agreed upon.
- Lack of structure: new companies may not have an appropriate organizational structure to handle large projects or a high volume of work.

Recognizing the importance of hiring companies with the highest level of market experience possible, this FPV represents 5% of the overall score and is divided as follows:

- Excellent: ≥10 years;
- Good: ≥8 years;
- Regular: ≥5 years;
- Poor: ≥3 years;
- Very Poor < 3 years.

Table 11 summarizes the ratings for each company in FPV 5—Time in the market.

**Table 11.** Scores on FPV 5—Time in the market.

| Company | Scores on FPV 5 |
|---|---|
| Company A | 0 |
| Company B | 10 |
| Company C | 10 |
| Company D | 10 |
| Company E | 5 |
| Company F | 10 |
| Company G | 7.5 |
| Company H | 0 |

In the context of this FPV, companies B, C, D, and F achieved the highest ratings, standing out for having ten years or more of experience in the RFT sector, followed by companies E and G, with eight and five years of operation, respectively. On the other hand, companies A and H performed the worst as they have fewer than three years of experience in the field.

### 4.1.6. Analysis of FPV 6—Sustainability

Sustainability refers to the capacity to meet the needs of present generations without compromising the ability of future generations to meet their own needs. This definition was proposed by the World Commission on Environment and Development of the United Nations (UN) in 1987, in the report "Our Common Future." Sustainability is a concept that involves actions and policies aimed at promoting balanced and integrated economic, social, and environmental development to ensure a sustainable future for the next generations [69].

Considering the relevance of sustainability in business operations, this FPV represents 5% of the model and was created based on binary ratings (0 or 10), depending on whether the evaluated companies have adopted sustainability actions or not.

In the realm of cargo transportation, adopting measures to reduce pollutant emissions and resource usage, improve activity performance, reduce operational costs, optimize processes, and enhance the quality of life for employees are actions contributing to sustainability. To receive a rating in this FPV, companies should have at least one practice that demonstrates a commitment to sustainability [70].

Table 12 gathers the ratings for each company in the FPV 6—Sustainability:

**Table 12.** Scores on FPV 6—Sustainability.

| Company | Scores on FPV 6 |
|---|---|
| Company A | 0 |
| Company B | 10 |
| Company C | 10 |
| Company D | 10 |
| Company E | 0 |
| Company F | 10 |
| Company G | 0 |
| Company H | 0 |

Among the evaluated companies, exactly four of them were found to promote sustainability initiatives. These companies are B, C, D, and F. On the other hand, the remaining companies stated that they have not implemented any activities of this nature thus far.

### 4.1.7. Analysis of the Global Scores

Having gone through the analysis and compilation of scores for all the EPVs and FPVs that make up the decision model, it remains to announce which of the eight evaluated companies obtained the highest overall score and, consequently, would be the most recommended company for transporting three tons of soybeans on the São Paulo (SP) to Belo Horizonte (BH) route. The summary of results can be found in Table 13.

**Table 13.** Global evaluation per company.

| Company | Global Scores |
|---|---|
| Company A | 4.62 |
| Company B | 3.74 |
| Company C | 7.27 |
| Company D | 6.69 |
| Company E | 5.98 |
| Company F | 2.31 |
| Company G | 6.48 |
| Company H | 5.98 |

Company C was the highest rated among the interviewed companies, according to the decision model used, which took into account criteria related to service price, delivery time, fleet age, location services, contact channels, company reputation, market experience, online reputation, and sustainability. The choice of company C to carry out the cargo

transportation highlights the importance of considering criteria beyond price that bring the highest possible quality to the service, reducing risks and losses for the contractor and the decision model user.

For a clearer view, Table 14 lists all scores obtained by company C throughout this field research.

**Table 14.** Results of the assessments for the criteria, sub-criteria, and global score for company C.

| FPVs | EPVs | EPV Score | EPV Weights | FPV Score | FPV Weights |
|---|---|---|---|---|---|
| FPV 1 | - | - | - | 7.5 | 55% |
| FPV 2 | 2.1<br>2.2<br>2.4 | 2.5<br>7.5<br>10 | 75%<br>15%<br>10% | 4 | 22.5% |
| FPV 3 | 3.1<br>3.2<br>3.3 | 10<br>10<br>10 | 33.33%<br>33.33%<br>33.33% | 10 | 7.5% |
| FPV 4 | - | - | - | 10 | 5% |
| FPV 5 | - | - | - | 10 | 5% |
| FPV 6 | - | - | - | 10 | 5% |
| Global Score | | | 7.275 | | |

When considering the transportation of three tons of soybeans on the São Paulo (SP) to Belo Horizonte (BH) route, company C emerges as the most suitable and secure choice. Even with the second-best price (equal to or less than USD 589) and a delivery time of 4 business days, the company received perfect scores in significant criteria, such as cargo location service, efficient communication channels, a strong reputation in the market, a long history of operation, and a commitment to sustainability. These factors highlight that company C possesses a solid structure and is dedicated to providing high-quality service to its clients while mitigating relevant risks associated with cargo transportation. Therefore, considering the route and the type of cargo in question, company C is the safest and most appropriate choice for transporting the three tons of soybeans.

### 4.2. Application of Sensitivity Analysis

It is important to note that all the companies in the field survey used the same model; therefore, a single sensitivity analysis should be sufficient to assess the reliability of this model. Company C, with the highest overall rating, was selected.

After calculating the criteria evaluations and the overall evaluation, a sensitivity analysis was conducted based on the parameters described in Section 3.9. This analysis was carried out by modifying the substitution rates of the criteria by +10% and −10%. After each modification, the overall evaluation was recalculated to determine if a slight change in the substitution rate would result in a significant change in the overall evaluation. The results are summarized in Table 15.

**Table 15.** Sensitivity analysis—Company C.

| FPV | Original Weights | Original Scores | Weights (+10%) | Recalculated Scores | Variations | Weights (−10%) | Recalculated Scores | Variations |
|---|---|---|---|---|---|---|---|---|
| FPV 1 | 55% | 7.275 | 60.50% | 7.309 | 0.47% | 49.50% | 7.244 | −0.42% |
| FPV 2 | 22.5% | 7.275 | 24.75% | 7.197 | −1.07% | 20.25% | 7.357 | 1.13% |
| FPV 3 | 7.5% | 7.275 | 8.25% | 7.317 | 0.58% | 6.75% | 7.243 | −0.44% |
| FPV 4 | 5% | 7.275 | 5.50% | 7.298 | 0.32% | 4.50% | 7.251 | −0.32% |
| FPV 5 | 5% | 7.275 | 5.50% | 7.298 | 0.32% | 4.50% | 7.251 | −0.32% |
| FPV 6 | 5% | 7.275 | 5.50% | 7.298 | 0.32% | 4.50% | 7.251 | −0.32% |

Considering that the variations in the overall evaluation scores ranged from a maximum of 1.13% to a minimum of −1.07%, we can infer that the constructed model is robust and the obtained evaluations are reliable.

## 5. Conclusions

Based on the knowledge presented in the literature review and with the assistance of industry professionals, this work constructed an efficient decision model that proved reliable during a simulated hiring process. During the sensitivity analysis, the model demonstrated its robustness by showing variations slightly greater than 1%, reinforcing its reliability. This decision model was able to support the hiring of a road freight transport company among eight competitors, providing a solid basis for decision-making.

The selected company C obtained an overall rating of 7.27 on a scale from 0 to 10. It is noteworthy that it received a score of 7.5 for the total freight price and 7.7 for inherent transportation aspects, including delivery time, fleet age, and cargo location service. Moreover, it received the highest rating in all other evaluated factors, such as customer service channels, reputation, years of market presence, and sustainability.

Although other competitors achieved scores close to company C, including in crucial aspects like total freight price and delivery time, which represent some of the heaviest weights in the model, it is believed that choosing company C is the most accurate decision since it presents the lowest risk during cargo transportation compared to all the others.

One of the main limitations of this research was the low number of participating companies. Although the number of participants was sufficient to test the decision model, it would be enriching to obtain a larger number of evaluations.

Regarding the contributions intended by this work, it is believed that this study provided both significant practical and theoretical contributions. In practical terms, the work developed a decision model that can be effectively applied in the selection of logistics operators for companies of various sizes using the road transportation mode. Furthermore, the study also contributed to theoretical advancement by integrating the literature on the use of multicriteria decision models in the selection of logistics operators using MACBETH.

It is recommended that this model be tested on other routes and in real hiring processes to ascertain if the defined criteria and their respective weights align with the hiring needs of the road freight transport market.

**Author Contributions:** E.C.M. designed and performed the research. A.S.J. reviewed the assessment and the methodology. All authors have read and agreed to the published version of the manuscript.

**Funding:** This research received no external funding.

**Institutional Review Board Statement:** Not applicable.

**Informed Consent Statement:** Informed consent was obtained from all subjects involved in the study.

**Data Availability Statement:** Data are contained within the article.

**Conflicts of Interest:** The authors declare no conflicts of interest.

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
