# Peer review of "Multicriteria Model to Support the Hiring of Road Freight Transport Services in Brazil"

_sustainability, doi:10.3390/su16093804_

Round 1
Reviewer 1 Report
Comments and Suggestions for Authors The study focused on developing a decision model for supporting the hiring of road freight transport services in Brazil, utilizing two distinct techniques.
I propose excluding the extensive literature review; the paper's key information revolves around the discussion of the Brazilian Transportation Matrix. Note that Chart 3 is in Portuguese.
Including a literature review in this type of publication is unnecessary.
For Figure 2, consider using a different figure or one with English information.
Regarding L433, elaborate on other models for creating decision models, especially those integrating artificial intelligence, to justify the specific choice of the two models.
Concerning L493, provide a scientific rationale for interviewing members from Cargill and BOSS Wood products, or consider suppressing the company names if no justification is given.
In section 4.1.1, change the currency to USD, and ensure the decimal separator is a point throughout the paper.
For L879, adequately cite or explain what RECLAME AQUI is.
In L1073, rephrase the sentence.
The paper contains valuable scientific information, and the authors successfully achieved their goals.
Please rephrase the conclusion. Some phrases do not make sense.
Author Response
Response to the feedbacks:
1- We appreciate the feedback and understand the perspective, but we believe that keeping this section serves as one of the supports for building the model, as well as providing context for the logistics scenario in Brazil, which is not commonly known to the international audience of the magazine. Chart 3 has been updated;
2- The only relevant information in Figure 2 is the route taken in the simulation, whereby the Portuguese information consists of the names of the cities involved, which do not require translation;
3- Several other MCDA models were added, and the choice of the MACBETH methodology and Swing Weights tool was justified;
4- Names omitted, as recommended;
5- Currency changed to USD;
6- Reclame Aqui is now properly introduced;
7- L1073 was one of the references.
We sincerely appreciate your help.

Reviewer 2 Report
Comments and Suggestions for Authors
The paper, titled "Multicriteria Model to Support the Hiring of Road Freight Transport Services in Brazil," aims to develop a decision model to aid in the selection of road freight transport services in Brazil using a Multicriteria Decision Analysis (MCDA) approach, specifically employing the Measuring Attractiveness by a Categorical Based Evaluation Technique (MACBETH) method.
The research addresses the lack of structured methods for the selection and hiring of carriers, which impacts logistics operations' efficiency and costs.
Through literature review and professional interviews, the study constructs a model designed to support logistics operators in the road transport sector, focusing on practical applicability for companies reliant on these services.
Key aspects covered include the importance of road freight in Brazil, the challenges of selecting carriers based on multiple criteria, and the proposal of a comprehensive evaluation model to facilitate decision-making processes.
The methodology adopted in the paper demonstrates novelty primarily through the application of the Measuring Attractiveness by a Categorical Based Evaluation Technique (MACBETH) method within the context of hiring road freight transport services in Brazil.
This approach is distinct because it leverages a structured multicriteria decision analysis (MCDA) framework to address the specific challenges faced in the Brazilian logistics sector, especially in selecting and hiring freight carriers.
The novelty lies in adapting and applying an existing decision-making framework to a complex, real-world problem, providing a tailored solution that considers the unique aspects of Brazilian road freight logistics.
By doing so, the paper contributes to the body of knowledge by offering a new perspective on solving logistics challenges through a well-established decision analysis technique, highlighting its versatility and applicability to industry-specific problems.
The manuscript provides a detailed analysis and discussion on the results obtained from the application of the Multicriteria Decision Analysis (MCDA) methodology, specifically through the MACBETH approach, for the selection and hiring of road freight transport services in Brazil.
However, without direct access to specific results sections and their interpretations, it's challenging to fully assess their acceptability from an external perspective.
Author Response
Firstly, we appreciate the valuable feedback! Although the results and interpretations are specified in the last two sections, given that this is a hiring simulation using the model, we believe that only its future use in real hiring situations will be able to provide the necessary support and external perspective.

Reviewer 3 Report
Comments and Suggestions for Authors
Dear authors,
your study is very interesting, well done and also a useful scientific work.
I am a bit sorry that you used almost exclusively Spanish/Portuguese language literature. In such works, it is really advisable to ensure the diversity of used resources from each page.
I have only two small comments.
1. finalize the methodology, devote yourself to such basic research methods as analysis, synthesis, deduction...
2. Expand the bibliographic apparatus with the works of other authors, such as:
Funta R. 2021. Automated Driving and Data Protection: Some Remarks on Fundamental Rights and Privacy. Krytyka prawa, 13(4), pp. 106–118, doi: 10.7206/kp.2080-1084.495
and
Kaššaj M, Peráček T. 2024. Sustainable Connectivity—Integration of Mobile Roaming, WiFi4EU and Smart City Concept in the European Union. Sustainability, 16 (2):788. https://doi.org/10.3390/su16020788
Since these works are related to the digital development and transformation of European cities, the autonomy of vehicles, with such additions you will not only expand the diversity of the sources used, but also add greater scientific value and benefit to your work. The work will become more complex. In addition, the authors also address the issue of methodology.
Author Response
We deeply appreciate the feedback! The work by Kaššaj M and Peráček T is now included in our references and has been extremely helpful in enhancing the methodology with factors such as theoretical data, analytical methods, comparison, and synthesis.

Reviewer 4 Report
Comments and Suggestions for Authors
Upon reviewing the manuscript, it is evident that the authors have undertaken a valuable effort to construct a decision model for the hiring of road freight transport services in Brazil. However, several areas require attention to strengthen the manuscript.
- The results should be clarified in the abstract.
- Mention the contributions and the novelty in the introduction.
- At the end of the introduction, please add the structure of the manuscript.
- Better connection of the paragraphs in the literature review section.
- Add a summary of the gaps at the end of the literature review section.
- I have big doubts on the credibility of the manuscript as the analysis is based on only three decision makers. Usually, there should be a lot more stakeholders from different groups (professionals, academics, etc.).
- The companies, even though identified anonymously using the letters A to H, should have more details provided about them to better connect the results from the first step with the companies' characteristics.
Acceptable
Author Response
We greatly appreciate the feedbacks on the work. It was extremely important for us to improve the mentioned sections.
Responses to the feedbacks:
- The successful creation of the model has been added to the abstract.
- The study's contribution is specified in the introduction in the following excerpt: "However, despite the availability of materials on both topics, no relevant studies were found that apply the MCDA/MACBETH methodology to the selection of suppliers for the road transport mode, which represents the main contribution of this work."
- Structure added in the last paragraph.
- The order of several paragraphs was changed, and the section as a whole was revised and is now more concise.
- Study gaps added.
- The model was based not only on the practical experience of the three decision-makers but also on the bibliography of the subject, which not only contextualizes the scenario of freight hiring in Brazil but also justifies the choice of elements that compose the model. We understand its limitation regarding practical proof, given that the work uses a simulation, but we hope that its future use in real situations will be able to assess its effectiveness in real hiring scenarios.
- The criteria used for selecting these companies were further specified in the results section.
Best regards,
Eduardo C. Moretto

Round 2
Reviewer 1 Report
Comments and Suggestions for Authors
The authors sufficiently addressed the reviewer's inquiries, integrating their suggestions into the article. Therefore, I recommend accepting the article.
Reviewer 4 Report
Comments and Suggestions for Authors
This version has addressed all my concern commented earlier.